# Neuromuscular Characteristics of Unilateral and Bilateral Maximal Voluntary Isometric Contractions following ACL Reconstruction

**DOI:** 10.3390/biology12091173

**Published:** 2023-08-26

**Authors:** Riccardo Di Giminiani, Stefano Marinelli, Stefano La Greca, Andrea Di Blasio, Massimo Angelozzi, Angelo Cacchio

**Affiliations:** 1Department of Biotechnological and Applied Clinical Sciences, University of L’Aquila, 67100 L’Aquila, Italy; stefano.marinelli@student.univaq.it (S.M.); stefano.lagreca@graduate.univaq.it (S.L.G.); 2Department of Medicine and Aging Sciences, ‘G. D’Annunzio’ University of Chieti-Pescara, 66013 Chieti, Italy; andrea.diblasio@unich.it; 3Department of Life, Health and Environmental Sciences, University of L’Aquila, 67100 L’Aquila, Italy; massimo.angelozzi@univaq.it (M.A.); angelo.cacchio@univaq.it (A.C.)

**Keywords:** strength, injury, functional recovery, EMG activity, co-activation index

## Abstract

**Simple Summary:**

Only the two-third of athletes who undergo anterior cruciate ligament reconstruction (ACLR) return to their pre-injury level and to sports participation. The timing for a safe return to sports participation plays a crucial role in reducing reinjury risk, which implies sensitive and reliability neuromechanical assessments to understand whether the deficit or alteration in motor control persists. The changes following ACLR are considered neurophysiological dysfunctions and not a simple peripheral musculoskeletal injury, and, consequently, the brain activation that influences bilateral lower extremity function may have occurred and the neuromechanical alterations could affect not only the operated leg but also the contralateral leg. Our study investigated the maximal voluntary isometric contractions synchronised with surface electromyographic (sEMG) activity of the thigh muscles during unilateral and bilateral knee extension in individuals with ACLR. The results showed that asymmetries between the two lower limbs were found only during bilateral exertions. Therefore, bilateral exertions are essential to underline neuromechanical alteration following ACLR. These findings could be helpful to define guidelines of expected longitudinal adaptations to reduce asymmetries and optimize functional recovery.

**Abstract:**

Despite the advancement of diagnostic surgical techniques in anterior cruciate ligament (ACL) reconstruction and rehabilitation protocols following ACL injury, only half of the athletes return to sports at a competitive level. A major concern is neuromechanical dysfunction, which occurs with injuries persisting in operated and non-operated legs following ACL rehabilitation. One of the criteria for a safe return to sports participation is based on the maximal voluntary isometric contraction (MVIC) performed unilaterally and a comparison between the ‘healthy knee’ and the ‘operated knee’. The present study aimed to investigate MVIC in athletes following ACL rehabilitation during open kinetic chain exercise performed unilaterally and bilateral exercises. Twenty subjects participated in the present investigation: 10 male athletes of regional–national level (skiers, rugby, soccer, and volleyball players) who were previously operated on one knee and received a complete rehabilitation protocol (for 6–9 months) were included in the ACL group (age: 23.4 ± 2.11 years; stature: 182.0 ± 9.9 cm; body mass: 78.6 ± 9.9 kg; body mass index: 23.7 ± 1.9 kg/m^2^), and 10 healthy male athletes formed the control group (CG: age: 24.0 ± 3.4 years; stature: 180.3 ± 10.7 cm; body mass: 74.9 ± 13.5 kg; body mass index: 22.8 ± 2.7 kg/m^2^). MVICs synchronised with electromyographic (EMG) activity (recorded on the vastus lateralis, vastus medialis, and biceps femoris muscles) were performed during unilateral and bilateral exertions. The rate of force development (RFD) and co-activation index (CI) were also calculated. The differences in the MVIC and RFD between the two legs within each group were not significant (*p* > 0.05). Vastus lateralis EMG activity during MVIC and biceps femoris EMG activity during RFD were significantly higher in the operated leg than those in the non-operated leg when exertion was performed bilaterally (*p* < 0.05). The CI was higher in the operated leg than that in the non-operated leg when exertion was performed bilaterally (*p* < 0.05). Vice versa, vastus medialis EMG activity during RFD was significantly higher in the right leg than that in the left leg when exertion was performed bilaterally (*p* < 0.05) in the CG. MVICs performed bilaterally represent a reliability modality for highlighting neuromechanical asymmetries. This bilateral exercise should be included in the criteria for a safe return to sports following ACL reconstruction.

## 1. Introduction

In athletes undergoing anterior cruciate ligament reconstruction (ACLR), a functional assessment following a comprehensive rehabilitation program is fundamental for a safe return to sports [1]. In the literature review, it was reported that in a rehabilitation program with a follow-up period of ≥24 months, the percentage of athletes who returned to their pre-injury level and to sports participation was approximately 62%. Additionally, younger athletes showed a high risk of reinjury, with a total second ACL reinjury rate of 15%, an ipsilateral reinjury rate of 7%, and a contralateral injury rate of 8% [2,3].

The timing for a safe return to sports plays a crucial role in reducing reinjury risk, which implies sensitive physiological and biomechanical assessments to understand whether the deficit or alteration in motor control persists. Quadriceps and hamstring strength assessment strategies can be performed under isokinetic and/or isometric conditions, which, in turn, are integrated with functional test batteries (i.e., single-hop, triple-hop, crossover) [4,5].

Isometric measurements can be performed during a closed kinetic chain (CKC) or an open kinetic chain (OKC) exercise––specifically, the use of CKC is justified for the early rehabilitation phase because it reduces the anterior-directed intersegmental forces and increases the compressive tibiofemoral forces (the stability of the knee joint)––whereas the OKC exercise is selected in the late phase of rehabilitation [6]. During isometric assessment, muscle torque can be determined by preserving the joint and anatomical structures of the limb that does not produce any displacement or external work (velocity is equal to zero) [1,5]. In addition, the isometric strength of the lower limb, when normalised to body mass, appears to be more predictive than the quadriceps limb symmetry index in defining self-reported function, which is evaluated using the International Knee Documentation Committee index in individuals undergoing ACLR [7].

During a maximal voluntary isometric contraction (MVIC), the neuromuscular system develops the highest strength values as the central nervous system (CNS) is able to recruit the greatest number of motor units, and several seconds are necessary to reach the peak [8], and vice versa. The dynamics of human movements are often characterised by fast limb movements that take place in a short time range and may not allow maximal strength to be reached [9]. Therefore, through the rate of force development (RFD), defined as the rate of that contractile force arises at the onset of contraction, it is possible to detect functional significance in fast movements that involve contraction times of 50–250 ms [9]. This indicates that following an ACLR, the RFD represents a useful adjunct outcome measure for the decision to return athletes to sports by quantifying the adequate muscle activity that occurs within a 30-to-70-ms window from the onset of joint loading to effectively protect the ACL from excessive forces [10,11,12]. 

In the management of ACL rehabilitation, exercises involving isometric contraction of the hamstring muscles do not strain the ACL ligament independently, but according to the knee position or the magnitude of the muscle contraction of the extensors [13,14]. The antagonistic role of the hamstring muscles to reduce the strain on the ACL and improve knee joint stability should be mediated by an arc reflex that modulates its activity relative to the overloaded or geometrical configuration of the ligament [15]. Hamstring coactivation during isometric knee extension is higher in both ACL reconstructed and ACL deficient patients than in individuals with no history or symptoms of knee joint disorders. From a functional point of view, the increased gain obtained by ACL–hamstring synergy acting through gamma system neurones would serve to increase hamstring muscle stiffness and knee joint stability [16].

However, in the latter studies, it was not clear whether the highest ACL–hamstring synergy following reconstruction was determined via unilateral or bilateral exertions. Typically, strength assessments of the quadriceps and hamstring functions are performed in a unilateral modality by comparing the operated leg (OL) and the non-operated leg (N-OL) [12]. In other words, alterations that affect the contralateral leg after ACL injury are not considered [17,18,19,20].

Recently, individuals with ACLR have shown several neural alterations, such as high bilateral corticospinal excitability (it is more difficult for the individual to activate those neurones to fully control the muscle) with a concomitant smaller motor-evoked potential in the injured limb (signifying that once an action potential is produced at the motor cortex, less of that signal will actually reach the muscle in individuals with ACLR) using transcranial magnetic stimulation [21]. The latter study underlines that ACLR or ACL deficiency should be considered as a neurophysiological dysfunction and not a simple peripheral musculoskeletal injury. Specifically, mechanoreceptor impairments due to ACL rupture generate modifications of the ascending afferent pathway towards the CNS, which, in turn, is capable of eliciting alterations in CNS organisation and function and determining a different activation pattern of the brain [22]. In addition, rehabilitation programs could affect this central reorganisation, and the execution of several unilateral tasks may reduce ipsilateral motor cortex activation [23], as bilateral activation seems to be a feature of lower extremity movements regardless of the joint and complexity [24]. Bilateral motor tasks show greater activation in cortical, cerebellar, and subcortical regions than unilateral tasks, although the force exerted during the unilateral task requires twice the force levels [25]. These findings provide evidence that the ‘healthy’ knee may not serve as a reference to make comparisons of the functional status of the ‘operated knee’ as the changes in brain activation that influence bilateral lower extremity function may have occurred [23,26]. Therefore, asymmetries could be found between the two lower limbs during unilateral and bilateral tasks.

The present study aimed to investigate maximal voluntary contractions (isometric) synchronised with surface electromyographic (sEMG) activity of the thigh muscles during unilateral and bilateral knee extension in individuals who had undergone ACLR. Additionally, the RFD and co-activation index (CI index between the knee extensors and knee flexors) were determined. Based on the previous literature, we hypothesised asymmetries in the neuromuscular activation of the quadriceps muscles, hamstring muscles, and co-activation of both muscles in the operated knee during unilateral and bilateral exertions.

## 2. Materials and Methods

### 2.1. Experimental Procedures and Participants

Twenty participants took part in this cross-sectional study with repeated measures by using a non-probability sampling (convenience sampling) [27,28]: ten male athletes of regional-national level (skiers, rugby, soccer, and volleyball players) who had previously been operated on one leg and received a complete rehabilitation protocol (6–9 months) formed the ACL group, whereas ten healthy male athletes matched to ACL participants were included in the CG. In Table 1, the participants’ characteristics are reported. For the ACL group, the inclusion criteria were as follows: complete functional recovery with a time lag of at least 6 months and complete return to sports practice. The rehabilitation program was conducted in a center of the national health system (NHS) affiliated with our university and supervised according to the indications of Beynnon et al. [29]. For the CG the inclusion criteria, there were no previous ligament or meniscus injuries. The exclusion criteria for both groups were as follows: actual or a history of skeletal muscular or nervous injuries, neuromuscular system pathologies, herniated disks, arrhythmias, epilepsies, and comorbidities with other disturbances. The assessment was conducted at the Biomechanics Laboratory of the Department of Applied Clinical Sciences and Biotechnology of the University of L’Aquila and the Internal Review Board approved this study (Prot. n° 33/2022). Before the intervention, each participant signed an informed consent form. Participants visited the laboratory on two different occasions: on the first testing day, they familiarised themselves with the experimental procedures by performing a series of MVICs in unilateral (operated and non-operated limbs) and bilateral exertions [30], whereas in the second testing session, separated by at least 48 h [31], they underwent the experimental session, which included MVICs and sEMG activity recorded in the leg muscles.

### 2.2. MVICs and sEMG Activity Measurements

In the second testing session, each participant began with 20 min of warm-up (general and specific phase to stimulate cardiovascular functions and the neuromuscular system) before performing MVICs in the unilateral (operated and non-operated limbs) and bilateral exertions at the leg extension equipped with a strain gauge (Muscle-Lab, Bosco-System, Ergotest Innovation, Stathelle, Norway). The general warm-up included running on a treadmill for approximately 10 min at a constant speed of 7 km/h and 3 min of static stretching exercises that mainly involved the lower limbs. In the specific warm-up (10 min), the participants performed several squat exercises in eccentric-concentric and isometric conditions. The knee angle bent at 120° was measured using an electrogoniometer connected to Muscle-Lab software (Muscle-Lab, Bosco-System, Ergotest Innovation, Stathelle, Norway). The sitting position on the machine was regulated to align the knee joint centre with the centre rotation of the lever of the machine. In addition, the length of the lever was regulated relative to the point of contact between the ankle and position. The participants wore a lumbar belt to minimise hip movements, and their hands gripped the lateral support of the machine in order to better stabilise the body position during the maximal isometric tasks. During MVICs, each participant performed a maximum of three attempts (separated by a minute of rest) [32,33] in the unilateral (right - left) and bilateral modalities (right + left), and the best value of the maximal isometric force in each task was retained for analysis. The participants were instructed to contract their leg muscles as hard and fast as possible, and strong verbal encouragement was provided during each trial by an experimenter (the same person) who said to the participants ‘push, push, push’ during the task execution. The start was given at each participant, and the participant continued the task without a time constraint and ended when the force–time traces reached a plateau. The force–time histories were monitored in real-time through Muscle-Lab software (Muscle-Lab, Bosco-System, Ergotest Innovation, Stathelle, Norway) [34]. The sEMG was recorded for the vastus lateralis (VL), rectus femoris (RF), vastus medialis (VM), and biceps femoris (BF) of both thigh muscles, and it was synchronised with the MVICs using Muscle-Lab software. Prior to the placement of the electrodes for the sEMG recording, skin shaves and cleansing (ethanol) were performed to minimise the impedance (<5 kΩ). The EMG electrodes and cables were secured with elastic bands (Vetrap, 3M Italia, Pioltello, Italy) to prevent motion artefacts. The root mean square of the sEMG activity (sEMG_RMS_) was determined using triode electrodes (T3402 M, nickel-plated brass, electrode diameter = 1 cm, interelectrode distance = 2 cm; Thought Technology Ltd., Montreal, QC, Canada). The electrodes were placed in the previously mentioned muscles in accordance with the SENIAM (www.seniam.org, accessed on 1 January 2022) guidelines for the noninvasive assessment of sEMG [35]. The sEMG detection technique entailed full-wave true RMS conversion of the signal from the preamplifier with a sampling frequency of 100 Hz (sampling of the converted signal). The size of the averaging window was 100 ms (averaging is analogous) with a resolution of a 16 bit A/D converter. The EMG preamplifier characteristics were as follows: voltage supply, ±5 VDC; input impedance, 2 GΩ; common mode rejection rate, 100 dB; gain at 100 Hz, 500; 3 dB low-cut frequency, 8 Hz; and 3 dB high-cut frequency, 1.2 kHz (Muscle-Lab, Bosco-System, Ergotest Innovation, Stathelle, Norway).

### 2.3. Data Analysis

The MVIC was identified in the force–time relationship recorded using Muscle-Lab software (Ergotest Innovation, Porsgrun, Norway). A window of 400 ms around the peak of the MVIC (200 ms before and 200 ms after) was used to compute the sEMG_RMS_ (mV) of the extensor (VL, VM, and RF) and flexor (BF) muscles during unilateral (operated and non-operated limbs) and bilateral exertions [36] (Figure 1). 

The RFD was determined in the initial portion of the force–time relationship measured during MVICs [9], and the time interval used to calculate the RFD was 0.200 ms relative to the onset of MVICs [9]. In this window, the sEMG_RMS_ (mV) of the muscles involved were calculated (Figure 1). The quadricep muscle force (*F_m_*) was calculated by performing a static analysis as follows [37]: $$\sum \tau_{K}=0\;\tau_{m\;}+\tau_{e\;}=0\;(b_{m}\times F_{m})+(b_{e}\times F_{e})=0\;\left(b_{m}\times F_{m})=-(b_{e\;}\times F_{e}\right)\;F_{m}=\frac{{-b}_{e\;}\times F_{e}}{b_{m}}\;[N];$$

On point *K* (the knee joint), two significant torques act during isometric leg extension (static condition): the muscle torque (*τ_m_*) given by the vectorial product of the muscle moment arm (*b_m_*) multiplied the quadriceps muscle force (*F_m_*), and the external torque (*τ_e_*) given by the external lever arm (*b_e_*) multiplied the external force (*F_e_*). 

The *F_e_* was measured as described previously, whereas the length of *b_e_* was measured using a camera with markers placed on the lateral femoral condyle and on the lateral malleolus of the tibia; subsequently, the analysis [38] was performed through a Kinovea software (version 0.8.15, Kinovea Open Source Project, www.kinovea.org, accessed on 1 January 2021). The length of quadricep lever arm (*b_m_*) was calculated using the equation described by Hosseinzadeh et al. [39] (Figure 2). 

The CI index between the knee extensors and the flexors was determined using the following equation [40,41,42]:$$CI=\frac{\left[\sum_{i=1}^{n}\frac{lower\;EMGi}{higher\;EMGi}\;\left(lower\;EMGi+higher\;EMGi\right)\right]}{n}$$
where *i* is the sample number and *n* is the number of data samples in the interval. A lower *EMGi* is the minimum value of the muscle during electromyography, whereas a higher *EMGi* is the maximum value of the muscle. Clearly, this method does not consider whether a muscle acts as an agonist or antagonist during leg extension. 

### 2.4. Statistical Analysis

The data were not normally distributed, as revealed by the Shapiro–Wilk’s W test; therefore, a non-parametric statistical procedure was used. Differences in several variables between the operated and non-operated legs were tested using the Wilcoxon signed-rank test/two-tailed test, and the *p*-values were computed using an exact method. The analyses were performed using XLSTAT (Statistical and Data Analysis Solution, Addinsoft, New York, NY, USA 2022; https://www.xlstat.com, accessed on 1 March 2021). Statistical significance was set at *p* ≤ 0.05, and the meaningfulness of significant outcomes was estimated by calculating the ES of Cohen. The intra-class correlation coefficient (ICC, 95% confidence limit, lower confidence limit-upper confidence limit) of the measured variables was quantified using test-retest reliability (http://www.sportsci.org, accessed on 20 July 2023). ICC values < 0.50 are defined as ‘poor’, those from 0.50 to 0.69 are defined as ‘moderate’, those from 0.70 to 0.89 are defined as ‘high’, and those >0.90 are defined as ‘excellent’ [43].

## 3. Results

### 3.1. Reliability of the Measurements

The ICCs 95% confidence limit, and lower confidence limit–upper confidence limit of the measured variables ranged from ‘high’ (0.72) to ‘excellent’ (0.90) in ACL and CG. (Appendix A). 

### 3.2. MVIC and RFD

The muscle force exerted during MVIC between the two legs was not significant within the ACL (OL vs. N-OL) and CG (Right vs. Left) (*p* > 0.05) (Figure 3A–C). The muscle force during bilateral was higher than during unilateral exercise in ACL (*p* < 0.05; ES = 0.57–0.69) (Figure 3A) and CG (*p* < 0.05; ES = 1.01–1.02) (Figure 3C). 

The RTD did not show any significant differences between unilateral and bilateral exercises within each group (*p* > 0.05) (Figure 3B–D).

### 3.3. sEMG_RMS_ during MVIC and RFD

The VL sEMG_RMS_ activity during MVIC was significantly higher in the OL than that in the N-OL when execution was performed bilaterally (*p* < 0.05; ES = 1.35) (Figure 4A. Within the CG, any significant differences were found (*p* > 0.05) (Figure 4B). 

During bilateral execution, the BF sEMG_RMS_ activity related to RFD also showed a higher activation in the OL than that in the N-OL (*p* < 0.05; ES = 0.49) (Figure 5B). On the contrary, the comparison between the right and the left leg was not significant in the CG (*p* > 0.05) (Figure 5D). Regardless, the VM sEMG_RMS_ activity in the right leg (dominant leg) was higher than that of left leg (*p* < 0.05; ES = 0.60) (Figure 5C) but not within the ACL group (Figure 5A). 

### 3.4. CI Index

The CI indexes of the VL-BF, RF-BF, and VM-BF muscles in the OL were higher than that of the N-OL when the exertion was performed bilaterally (*p* < 0.05) and not unilaterally (*p* > 0.05). The effect size of the CI between the VL and BF (ES = 0.58) was higher than that between the RF and BF (ES = 0.33) and the VM and BF (ES = 0.43) (Figure 6).

## 4. Discussion

The present study clearly indicated that the neuromuscular activation of agonists, antagonists, and co-activation of both was significantly higher in the OL leg than that in the N-OL leg during bilateral but not during unilateral exertion; however, the two legs did not exhibit differences in MVIC and RFD. The results of our study confirmed an altered neuromuscular strategy in hamstring activation, as shown in other studies [15,16,20,44]. These asymmetries were not found in CG, and, conversely, the VM showed a higher activation in the dominant leg than in the non-dominant leg when the exertion was performed bilaterally. Similarly to the ACL group, the two legs did not show any significant differences during unilateral exertion.

In the ACL group, the hamstring activation is protective as it dynamically stabilizes the reconstructed limb, [45] assisting the ligament when it is overloaded by the anterior tibial translation during the action of knee extensors [46], and this neuromuscular behaviour determines an altered coordination in the thigh muscles [44] through gamma loop dysfunction of quadriceps femoris [47] and a reduced neural drive to the vastii muscles (which could reflect a reduced descending command and/or an increased inhibitory afferent feedback inputs to motoneurons) [48]. From a mechanical point of view, greater hamstring activation determines an increased compressive force on the knee and the lesser knee range of motion during gait, and long-term joint health following highly repetitive activities of daily living or sports competition could be compromised as this strategy is associated with cartilage degeneration, knee osteoarthritis, and reinjury risk [49]. 

However, in the literature, the higher hamstring activation in the ‘operated knee’ than in the ‘healthy knee’ was accompanied by quadriceps inhibition in the injured knee or on both sides during the maximal isometric contraction during the leg extension [15,20,47,50,51]. On the contrary, our study showed a higher VL (related to the peak of MVIC) and hamstring coactivation ratio (determined between VL-BF, RF-BF, and VM-BF in the initial portion of the force–time relationship, with RFD calculated with a window of 0.200 ms relative to the onset of MVICs) in the injured side than that in the uninjured side. 

Different explanations for the disagreement in the results emerge from the comparison of the experimental design used in our study with those of others [15,20,47,50,51]. First, the ACL of the participants enrolled in the present investigation was reconstructed using the combined semitendinosus–gracilis tendon (hamstring), whereas in the studies cited above, the harvest of a bone–patellar tendon bone (BPTB) was involved. In this regard, Ito et al. [52] demonstrated that during gait, the involved limb of the BPTB graft group had a longer neuromechanical deficit (electromechanical delay) than that of the hamstring group in the VL and VM muscles after 2 years of rehabilitation. This indicates that the mechanical properties of the patellar tendon in the central region of the tendon (where the BPTB graft is harvested) are altered and may influence neuromuscular control during gait, which persisted even after athletes completed 2 y of testing and returned to sports competition. Similarly, Smith et al. [53] showed that athletes with BPTB grafts took 1.5 months longer to recover a full range of motion, maximal voluntary isometric contraction (leg extension), and explosive strength (single leg-hopping test) than those who received a hamstring tendon autograft. In addition, graft type interacts with sex during the first 12 months following ACLR: the maximal voluntary isometric contraction (leg extension) and limb symmetry index values in the involved limb of males with BPTB grafts were significantly lower than those of males with hamstring autografts, whereas no differences were found between graft sources among females [19]. 

Another novel aspect of our measures compared with previous methods is the ability to discern different neuromuscular functions in unilateral and bilateral exertions. In contrast to previous studies, in which the paradigm for a safe return to sports was represented by a similar neuromuscular pattern of the two knees by comparing them during unilateral exertion [54,55], our findings provide evidence that alterations in the activation of the agonist (VL) and antagonist (BF) muscles in the operated knee emerge significantly only during bilateral exertion. 

Several movements in daily life are performed by bilateral exertion, which requires the activation of more brain areas (i.e., the cortical, subcortical, and cerebellum) and several motor regions with higher activation than unilateral exertion [25]. The extra neural activity of subcortical areas and the cerebellum is necessary to coordinate simultaneous bilateral leg movements and stabilise the body [25]. 

In contrast, participants with ACL deficiency show reduced activation of several sensorimotor cortical areas [22]. This pattern of whole-brain activity seems to be generated by modifications of the ascending afferents owing to functional impairments in the mechanoreceptors of the ACL [56]. Despite unilateral injuries, joint dysfunction is bilateral in nature [47,56]; therefore, it can be better identified via bilateral movements. Thus, many cortical resources may be required to stabilise the operated knee joint when the task is characterised by coordinated movements during bilateral exertion. 

This neuromechanical condition could also explain why the biceps femoris EMG activities relative to MVICs and RFD in our study showed higher reliability in bilateral than in unilateral exertion. Bilateral exertion showed the highest ICC values in the operated leg, 0.94 and 0.89 for EMG relative to MVICs and EMG relative to RFD, respectively, while during unilateral exertion, the ICC values were 0.94 and 0.72 (Appendix A). These results are in line with those reported by Pérez-Castilla et al. [57], in which bilateral counter movement jumps provided more reliable measures of interlimb asymmetry in healthy basketball players. Similarly, Cuthbert et al. [58] reported that bilateral exercise is more reliable for identifying interlimb asymmetry and assessing chronic changes in hamstring strength within and between sessions than unilateral exercise. Additionally, the unfamiliar nature of a unilateral leg extension may have altered the neural strategy, owing to the interlimb coordination accuracy required by many daily activities [59,60]. Further investigations are required to provide insight into the underlying central and/or peripheral neuromechanical mechanisms involved in the two-exertion modality.

The participant of CG showed asymmetries in VM activation between dominant and non-dominant during bilateral execution but not during unilateral execution. De Souza et al. [61] have highlighted that during isometric contraction, the VM shows a higher firing rate activation and a greater stability to patellar tracking than the VL; in this way, the delay of VL activation is able to promote a force production more efficiently as the knee joint has been stabilised by the VM activation. Venturelli et al. [62] reported that VL EMG activity decreased with a concomitant increase in the mechanical efficiency in the dominant limb. In any case, the EMG activity was recorded in the VL during unilateral exertion. 

Other studies that used isometric contractions found a significant difference in quadriceps MVIC torque between the dominant and non-dominant limb during unilateral exertion [63,64]. This difference could be explained by the type of sports practised by the participants, which could affect the results of the measurements. In fact, Rahnama et al. [65] reported differences between the dominant and non-dominant limb during maximal isokinetic unilateral contraction in football players. The authors stated that the sport-specific demands (i.e., kicking, weight balance during kick, jumping, and sprinting) on the neuromuscular system of the player could induce specific adaptations underlying these changes. The latter study is supported by a recent investigation [66] in which athletes who practise resistance training, display a greatest proportion of myosin heavy-chain fibres I (MHC-I) in the dominant leg, whereas the myosin heavy-chain fibres IIa (MHC-IIa) were highest in the non-dominant leg. This suggests that peripheral adaptations could be caused by a complex of physiological multifactorial mechanisms (i.e., neurological, elastic component, muscle quality, and single myofiber size) [66].

In addition, our results in ACL and CG have highlighted differences in neuromuscular activation only during bilateral exertion. They could be probably explained by the bipedal nature of the lower limbs, which are usually recruited bilaterally. In fact, we observed that bilateral execution exhibits greater reliability than unilateral execution. For example, the ICC of BF sEMG activity, relative to the RFD, is equal to 0.94 (0.80–0.98) during bilateral exertion and 0.72 (0.39–0.90) during unilateral exertion.

Although in line with other studies, a point of concern of our investigation regards the relatively small sample size in each group (n = 10). Post hoc analysis of VL activation in the OL during bilateral exertion showed the highest power (1-β err prob = 0.93), vice versa for the other significant differences the power was lowest (1-β err prob = 0.37–0.54). 

## 5. Conclusions

Our results suggest that bilateral movements during leg extensions represent a reliable modality to highlight neuromechanical asymmetries following ACLR. Bilateral exertion is also essential to underline differences in neuromuscular activation between the dominant and non-dominant leg in healthy athletes, and they should therefore be included in the assessment for functional recovery and prevention. The extra neural activity required to coordinate bilateral movements could be useful as an ‘adaptive strategy’ in the rehabilitation or training process, considering its potential effects on neuroplasticity. The results of this investigation underline that bilateral exertion represents a reliable procedure for kinesiologists and therapists to monitor the rehabilitation or training process. In this way, helpful guidelines of expected longitudinal gains could be provided to reduce asymmetries and to optimize the adaptations.

## Figures and Tables

**Figure 1 biology-12-01173-f001:**
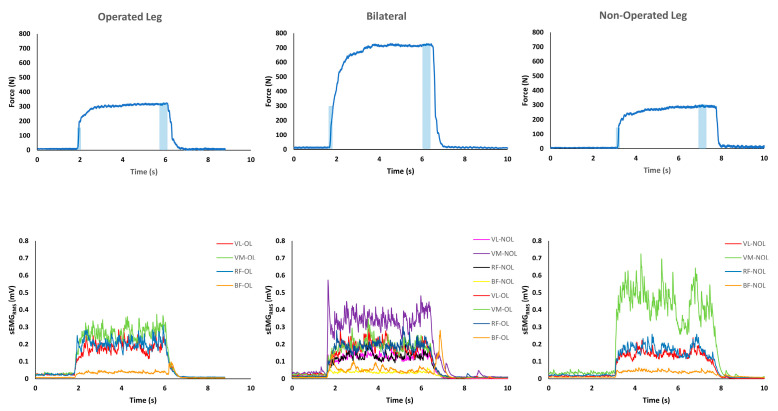
Representative example of force-time and surface electromyography root-mean-square (sEMG_RMS_)-time signals recorded during unilateral (operated leg, OL and non-operated leg, N-OL) and bilateral maximal voluntary isometric contractions (MVICs) during the leg extension. The sEMG_RMS_ activity was recorded in the vastus lateralis (VL), vastus medialis (VM), rectus femoris (RF), and biceps femoris (BF) muscles. The larger shaded areas represent the window (0.4 s) used to compute the synchronised muscles sEMG_RMS_ around the peak of MVICs. The rate of force development (RFD) and the synchronised sEMG_RMS_ were computed 0.20 s after the force exertion (the smallest shaded areas).

**Figure 2 biology-12-01173-f002:**
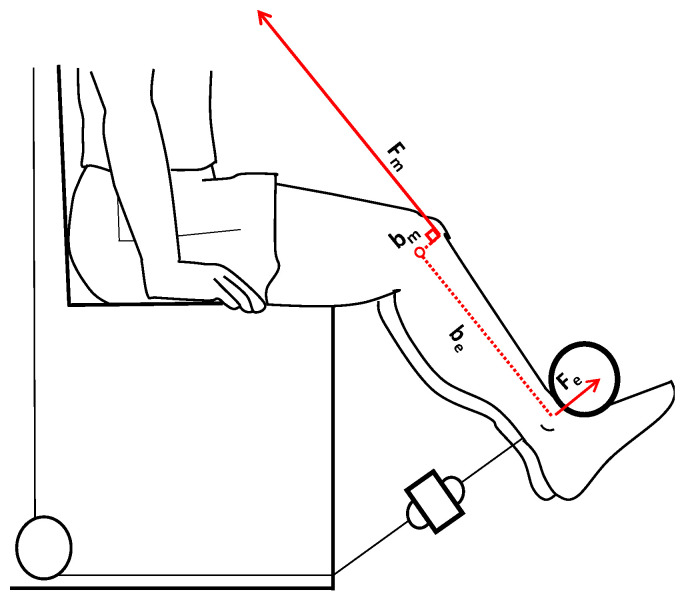
Schematic representation of quadricep muscle force calculation during isometric leg extension. The external force (*F_e_*) is measured by means a strain gauge. The muscle force (*F_m_*) is calculated. The external lever arm (*b_e_*) and the muscle moment arm (*b_m_*) are also indicated in the picture.

**Figure 3 biology-12-01173-f003:**
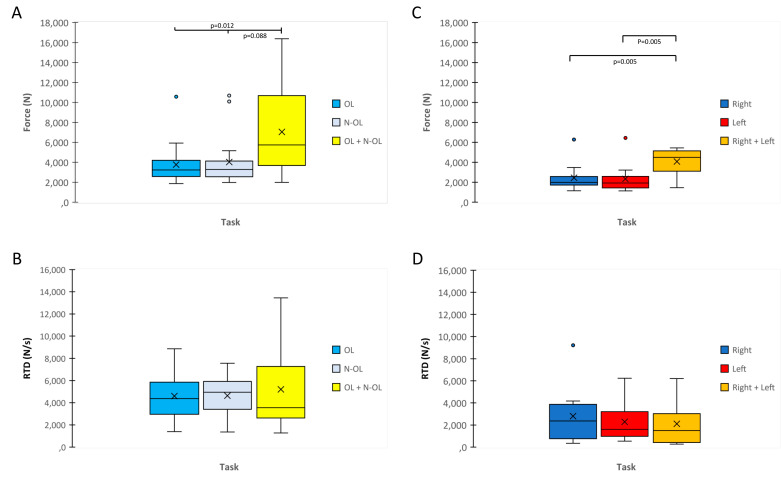
Maximal Voluntary Isometric Contraction (MVIC) and Rate of Force Development (RFD) in different tasks are reported with box plots for ACL (**A**,**B**) and CG (**C**,**D**). OL, Operated Leg; N-OL, Non-Operated Leg.

**Figure 4 biology-12-01173-f004:**
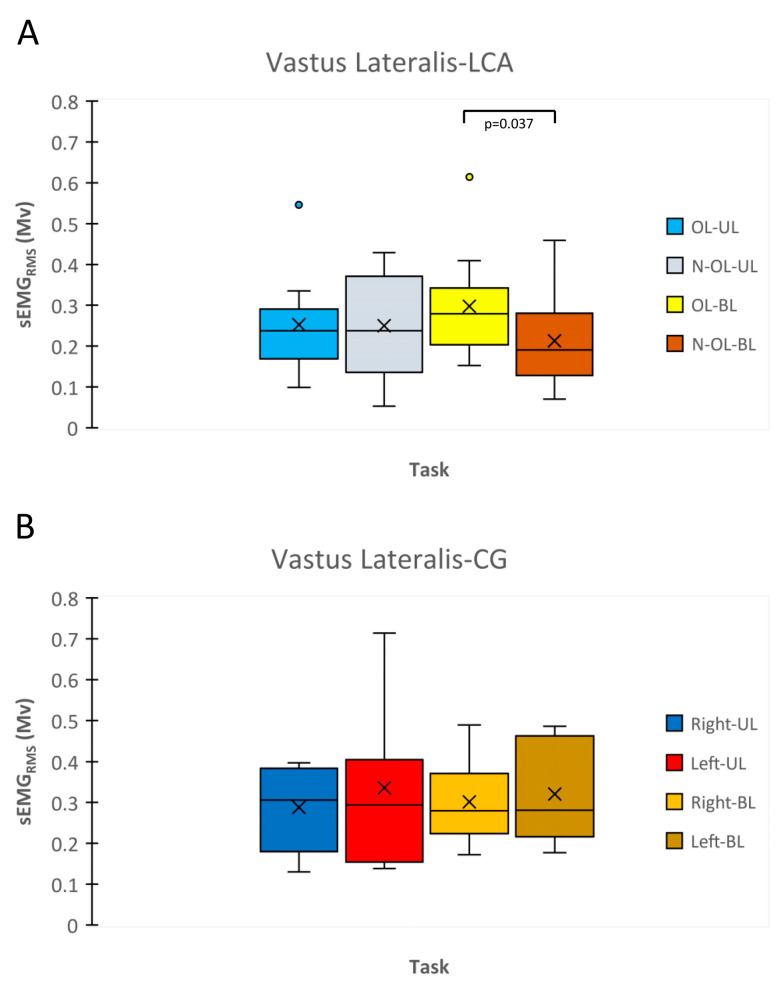
Surface electromyography root-mean-square (sEMG_RMS_) of vastus lateralis muscle relative to maximal voluntary isometric contraction (MVIC) is reported with box plots for ACL (**A**) and CG (**B**). OL, Operated Leg; N-OL, Non-Operated; UL, Unilateral; BL, Bilateral.

**Figure 5 biology-12-01173-f005:**
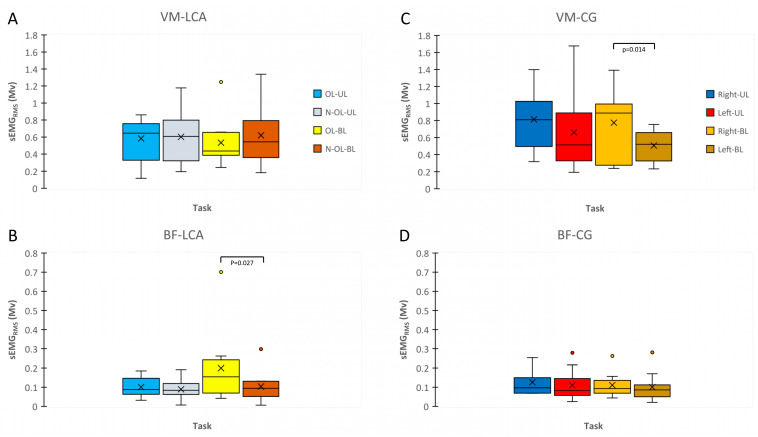
Surface electromyography root-mean-square (sEMG_RMS_) of vastus medialis muscle (VM) and biceps femoris (BF) relative to the rate of force development (RFD) is reported with box plots for ACL (**A**,**B**) and CG (**C**,**D**). OL, Operated Leg; N-OL, Non-Operated; UL, Unilateral; BL, Bilateral.

**Figure 6 biology-12-01173-f006:**
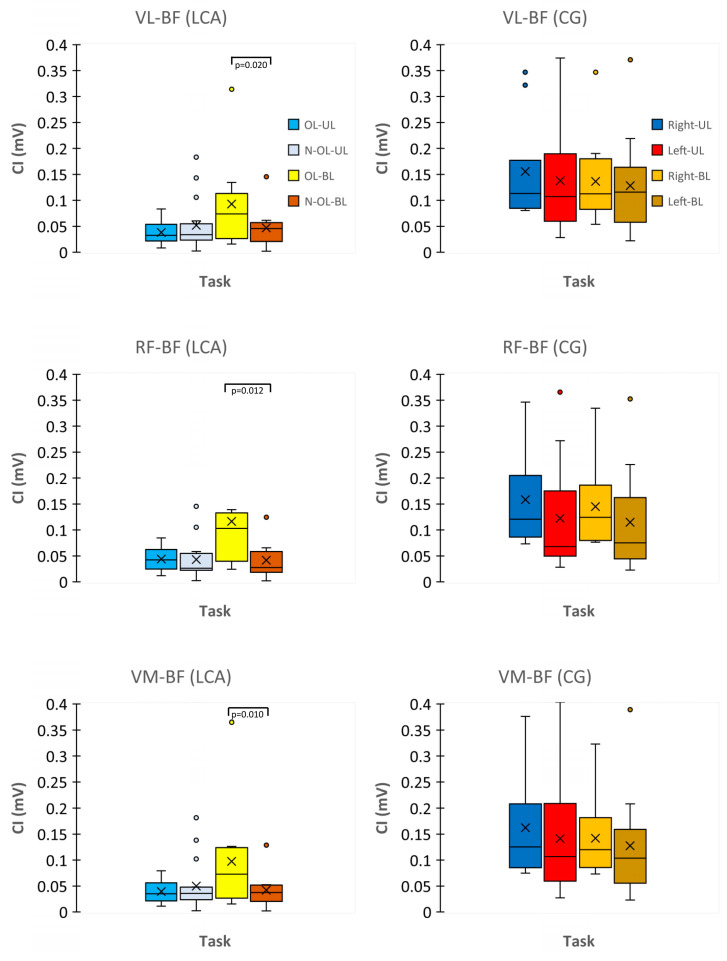
Coactivation Index (CI; sEMG_RMS_) of leg muscles (vastus lateralis-biceps femoris, VL-BF; rectus femoris-biceps femoris, RF-BF; vastus medialis-biceps femoris, VM-BF) relative to the rate of force development (RFD) is reported with box plots in ACL and CG. OL, Operated Leg; N-OL, Non-Operated Leg; UL, Unilateral; BL, Bilateral.

**Table 1 biology-12-01173-t001:** Participants’ Characteristics.

Variable	Group	
	ACL Group (*n* = 10)	Control Group (*n* = 10)
Age (Years)	23.4 ± 2.1	24.0 ± 3.4
Stature (cm)	182 ± 9.9	180.3 ± 10.7
Body Mass (Kg)	78.6 ± 9.9	74.9 ± 13.5
BMI (Kg/m^2^)	23.7 ± 1.9	22.8 ± 2.7
Sports level	*Competitive*	*Competitive*
Leg Dominance	*Right* (*n* = 9)/*Left* (*n* = 1)	*Right* (*n* = 9)/*Left* (*n* = 1)
Operated Leg	*Right* (*n* = 6)/*Left* (*n* = 4)	*NA*
Graft	*SGT* (*n* = 10)	*NA*
Event Distribution	*No-Contact Mechanism*	*NA*
Post Operative Period	*6 Months–2 Years*	*NA*

Abbreviations: body mass index (BMI), semitendinosus–gracilis tendon (SGT).

## Data Availability

The raw data supporting the present study will be made available by the authors without restriction.

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
