# Peer review of "Neuromuscular Characteristics of Unilateral and Bilateral Maximal Voluntary Isometric Contractions following ACL Reconstruction"

_biology, 2023, doi:10.3390/biology12091173_

Round 1

Reviewer 1 Report

REVIEW: Neuromuscular Characteristics of Unilateral and Bilateral Exercise during Maximal Voluntary Isometric Contractions Following ACL Reconstruction

GENERAL COMMENTS

The authors aimed to investigate Maximal Voluntary Isometric Contractions (MVIC) in athletes following ACL rehabilitation during open kinetic chain exercises performed unilaterally and bilaterally. They concluded that MVICs performed bilaterally represent a sensitive modality for highlighting neuromechanical dysfunctions and that bilateral exercise should be included in the criteria for a safe return to sports following ACL reconstruction.

This clear and concise experimental study addressed an important topic related to assessing potential limb asymmetries concerning the biomechanical properties of thigh muscles in subjects recovering after ACL. However, before accepting it for publishing, some issues should be addressed first. Throughout the article, there is confusion regarding using the term exercise. Namely, the authors explored the characteristics of Unilateral and Bilateral Maximal Voluntary Knee extensor single sustained contractions. Using the term exercise is unnecessary since they test muscle function while contracting maximally against unmovable resistance, which is correctly explained in the methods section. So, the authors (starting with the title) use unilateral and bilateral maximum voluntary isometric contractions of knee flexors and extensors instead of “exercise.” The title could be shortened to: “Neuromuscular Characteristics of Unilateral and Bilateral Maximum Voluntary Isometric Contractions of Knee Extensors Following ACL Reconstruction.” However, I do not insist on strictly following my recommendation (it’s just a suggestion to improve clarity). It could be sufficient to define what they mean by exercise in the article.  

SPECIFIC COMMENTS

INTRODUCTION

·        Please consider adding the following reference in the paragraph starting with Line 106:
Mirkov, D. M., Knezevic, O. M., Maffiuletti, N. A., Kadija, M., Nedeljkovic, A., & Jaric, S. (2017). Contralateral limb deficit after ACL-reconstruction: an analysis of early and late phase of the rate of force development. Journal of sports sciences, 35(5), 435-440.

·        Please clarify (be more specific) your hypothesis statement.

METHODS

·        Line 158: Please substitute force-time history with force-time trace (or signal).

·        Line 184: Please consider adding an Illustration of how the quadricep muscle force (Fm) has been assessed (calculated).

·        Since the Force was measured during Isometric contraction (no movement) at the fixed knee angle, please provide a rationale for using the described approach to assessing muscle force. Particularly add a description of how the muscle moment arm (mb) was measured (Since in the reference provided (Hosseinzadeh et al. 2020), the regression equation was used to assess the length of the muscle moment arm for a given knee angle). Did the MRI was used?

·        Line 200 At the end of the sentence starting at line 200 and ending at line 202, erase (bm) after reference.

·        Please consider placing the results of the operated and non-operated legs next to each other in Figure 1.

DISCUSSION

Please provide in more detail the potential benefits of the obtained finding for practitioners involved in assessing knee muscle function in patients recovering after knee injuries. 

Author Response

Author’s response

First, we would like to thank your comprehensive review of our manuscript. Your comments helped us to improve our study. We have modified the manuscript following your suggestions. Corrections and changes highlighted with yellow were made in the text.

Reviewer#1

SPECIFIC COMMENTS

INTRODUCTION

  • Please consider adding the following reference in the paragraph starting with Line 106:
    Mirkov, D. M., Knezevic, O. M., Maffiuletti, N. A., Kadija, M., Nedeljkovic, A., & Jaric, S. (2017). Contralateral limb deficit after ACL-reconstruction: an analysis of early and late phase of the rate of force development. Journal of sports sciences, 35(5), 435-440.
  • Please clarify (be more specific) your hypothesis statement.

Author’s response

  • We have included the reference.
  • We have clarified the research hypothesis of our study.

Reviewer#1

METHODS

  • Line 158: Please substitute force-time history with force-time trace (or signal).
  • Line 184: Please consider adding an Illustration of how the quadriceps muscle force (Fm) has been assessed (calculated).
  • Since the Force was measured during Isometric contraction (no movement) at the fixed knee angle, please provide a rationale for using the described approach to assessing muscle force. Particularly add a description of how the muscle moment arm (mb) was measured (Since in the reference provided (Hosseinzadeh et al. 2020), the regression equation was used to assess the length of the muscle moment arm for a given knee angle). Did the MRI was used?
  • Line 200 At the end of the sentence starting at line 200 and ending at line 202, erase (bm) after reference.
  • Please consider placing the results of the operated and non-operated legs next to each other in Figure 1

Author’s response

  • We have substituted “force-time history” with “force-time traces” in the text.
  • We have included a picture in the text.
  • We did not use MRI. We have estimated the muscle moment arm by using the equation of Hosseinzadeh et al. (2020).
  • Line 200. We have corrected in the text.
  • We have changed the figures in the text.

Reviewer#1

DISCUSSION

 Please provide in more detail the potential benefits of the obtained finding for practitioners involved in assessing knee muscle function in patients recovering after knee injuries. 

 Author’s response

  • We have included the potential benefits of the obtained finding in the conclusion section.

Reviewer 2 Report

The objective of this work is to describe the neuromuscular characteristics of the thigh musculature in individuals who have suffered an anterior cruciate ligament (ACL) rupture and have undergone reconstruction using hamstring muscle grafts.

The main strength of this study lies in the use of gold-standard assessment systems and the homogeneity of the selected hamstring muscle graft for surgical intervention. However, there are several limitations in the theoretical and methodological approach, as well as in the interpretation of results and conclusions.

In the introduction:

• L89-L108 provides specific background information on the central nervous system response following injury, but the research is limited to evaluating the peripheral nervous system response, thereby limiting the interpretation of results at the central nervous system level. What relevance do these paragraphs have in the study?

• The study objective is not sufficiently clear and does not align with the information presented in L89-L108. How do the authors intend to connect this information with their objective? What is the purpose of investigating "maximal voluntary contractions (isometric) synchronized with surface electromyographic (sEMG) activity of the thigh muscles during unilateral and bilateral knee extension in individuals who had undergone ACLR. Additionally, the RFD and co-activation index (CI index between the knee extensors and knee flexors) were determined"?

• The proposed hypothesis also does not seem to relate to the approach presented in L89-L108.

In the methods section, several aspects require attention:

• The study design does not allow for the identification of a clear control group, as the authors themselves state that the non-operated leg cannot be considered a pure control.

• Although the authors claim that all participants received a hamstring graft in the discussion section, it is important to add this information to the description of the selected participants.

• The participants' level and mode of sports activity are heterogeneous.

• The sample size is small. Could you clarify the calculation of the sample size estimation?

• What sampling technique was used?

• Males and females are included in the same group, with an unbalanced proportion. As stated by the authors in the discussion, neuromuscular control strategies appear to be different between males and females.

• The control of the rehabilitation process has not been described or controlled. Could it have affected the participants' neuromuscular recovery?

• The comparison and interpretation of raw electromyographic signals through voltage is questionable. Factors such as muscle volume and length can affect the signal magnitude, making it difficult to compare neuromuscular activation between different muscle groups and even between individuals.

• The recovery time between the familiarization session and the assessment session may have been insufficient (48 hours), considering that the average recovery time for neuromuscular efforts is 72 hours. Could you provide any results/evidence justifying this decision?

• The recovery time between repetitions may also be limited (1 minute). Could you clarify why this time was chosen or if a specific protocol was followed?

• L155-156 This explanation is repeated in lines 160-161. Please unify. • L171-178 Could you clarify why these parameters were chosen for signal filtering?

• The statistical analysis methods used do not seem appropriate. First, the multiple comparisons conducted could increase the probability of Type I error. Perhaps you can consider multivariate analysis techniques by transforming the data. Additionally, the mean and standard deviation are not the best statistics for describing the outcome of variables that do not follow a normal distribution. In these cases, you may consider using the median and interquartile range or other robust measures of central tendency.

• The link (http://www.sportsci.org/2007/simulatesamples.xls) is broken.

In the results section:

• L241. The mean lower limit of ICC is 0.66, not 0.63. In any case, the lower limit occasionally reaches an ICC of 0.10.

• Some of the lower limits of ICC seem to indicate poor result reproducibility.

• In section 3.2, the wording of the results is unclear. The figure used combines the representation of different variables and styles (bars vs. points+error bars). Considering the non-normal distribution of the data, it may be preferable to present the results in the form of a table or a figure representing box plots.

• Effect sizes should also be added in sections 3.2 and 3.3. Additionally, it should be considered to separate the graphs that present the comparisons between the operated and non-operated leg unilaterally and bilaterally, even if they are included in a single image. For example, graphical format A, graphical format B. Otherwise, it appears that the comparisons are made among all groups and tasks.

The discussion is conditioned by the methodological weaknesses previously described, so it is necessary to review it in future stages of the review process.

The conclusion is not clearly aligned with the problem, objective, and results of the study. The results do not allow for the verification of the sensitivity or reliability of the protocol used. The wording of the conclusion needs to be reformulated.

Author Response

Author’s response 

First, we would like to thank your comprehensive review of our manuscript. Your comments helped us to rethink our study and paper. It was very useful. As you can see, we accepted many of your suggestion, rewriting, including a control group and rephrasing the paper. However, in some cases we tried to give answers to your questions and to reason of our opinion. Thank you very much for your kind assistance in advance.

Corrections and changes highlighted with yellow were made in the text.

Reviewer#2 comment

In the introduction:

  • L89-L108 provides specific background information on the central nervous system response following injury, but the research is limited to evaluating the peripheral nervous system response, thereby limiting the interpretation of results at the central nervous system level. What relevance do these paragraphs have in the study?
  • The study objective is not sufficiently clear and does not align with the information presented in L89-L108. How do the authors intend to connect this information with their objective? What is the purpose of investigating "maximal voluntary contractions (isometric) synchronized with surface electromyographic (sEMG) activity of the thigh muscles during unilateral and bilateral knee extension in individuals who had undergone ACLR. Additionally, the RFD and co-activation index (CI index between the knee extensors and knee flexors) were determined"?
  • The proposed hypothesis also does not seem to relate to the approach presented in L89-L108.

Author’s response 

The hypothesis is developed not only on the approach in L89-L108. The entire narrative of the introductory paragraphs introduces to the necessary background information and explains the rationale behind the research hypothesis. The paragraph is a part of the rationale of our research hypothesis and underlines that ACLR or ACL deficiency should be considered neurophysiological dysfunction and not a simple peripheral musculoskeletal injury; it means that the ‘healthy’ knee may not serve as a reference to make comparisons of the functional status of the ‘operated knee’ as the changes in brain activation that influence bilateral lower extremity function may have occurred.

In the management of ACLR, for a safe return to sport participation, is suggested a simple comparison operated versus non-operated leg! Therefore, we propose assessing any asymmetries between the two lower limbs (during unilateral and bilateral tasks).

Our investigation is the first to considerer unilateral and bilateral tasks in ACLR. However, we are aware that further investigations will be needed to clarify the exact neuromechanical mechanisms responsible of our results.

Reviewer#2 comment

In the methods section, several aspects require attention:

  • The study design does not allow for the identification of a clear control group, as the authors themselves state that the non-operated leg cannot be considered a pure control.
  • Although the authors claim that all participants received a hamstring graft in the discussion section, it is important to add this information to the description of the selected participants.
  • The participants' level and mode of sports activity are heterogeneous.
  • The sample size is small. Could you clarify the calculation of the sample size estimation?
  • What sampling technique was used?
  • Males and females are included in the same group, with an unbalanced proportion. As stated by the authors in the discussion, neuromuscular control strategies appear to be different between males and females.
  • The control of the rehabilitation process has not been described or controlled. Could it have affected the participants' neuromuscular recovery?
  • The comparison and interpretation of raw electromyographic signals through voltage is questionable. Factors such as muscle volume and length can affect the signal magnitude, making it difficult to compare neuromuscular activation between different muscle groups and even between individuals.
  • The recovery time between the familiarization session and the assessment session may have been insufficient (48 hours), considering that the average recovery time for neuromuscular efforts is 72 hours. Could you provide any results/evidence justifying this decision?
  • The recovery time between repetitions may also be limited (1 minute). Could you clarify why this time was chosen or if a specific protocol was followed?
  • L155-156 This explanation is repeated in lines 160-161. Please unify.
  • L171-177. Could you clarify why these parameters were chosen for signal filtering?
  • The statistical analysis methods used do not seem appropriate. First, the multiple comparisons conducted could increase the probability of Type I error. Perhaps you can consider multivariate analysis techniques by transforming the data. Additionally, the mean and standard deviation are not the best statistics for describing the outcome of variables that do not follow a normal distribution. In these cases, you may consider using the median and interquartile range or other robust measures of central tendency.
  • The link (http://www.sportsci.org/2007/simulatesamples.xls) is broken.

Author’s response

  • We included a control group with characteristics to match the ACL group.
  • The participants’ characteristics are reported in Table 1.
  • The sample size estimation was appropriate for the experimental design. It was computed a priori for within group comparisons. We used G-Power (G*Power 3.1.9.4; Heinrich Heine-Dusseldorf University), setting the effect size (ES) values to those of previous studies and using the protocol for a power analysis (test attributes: Wilcoxon signed-rank test (matched pairs); ES=0.8, α = 0.05, power [1-β] = 0.90, sample size n = 20 participants. Now, in the revised version of our manuscript the ACL group was reduced to 10 participants to make it homogeneous (see points below). Anyway, we performed a post-hoc analysis and included the results in the limitations at the end of discussion section.
  • We used a non-probability sampling (convenience sampling) (Kish L., 1994; Thobane et al., 2022). We recruited the participants among the population of the region Abruzzo (near to our university) to supervise the rehabilitation process following the indications of Beynnon et al. (2005). We have included this information in the method section.
  • Now, the ACL group is formed only by male participants (Table 1).
  • The rehabilitation program was conducted in a center of the national health system (NHS) affiliated with our university and supervised according to the indications of Beynnon et al. (2005). We included this information in the text.
  • This is a known problem as electrode placement, fat, cell hydration and more, will affect the magnitude of the voltage. A common way to work around that is to ‘normalize’ the signal by recording the EMG signal magnitude at a known load or during MVC and use that as reference. Anyway, in the present study we made comparisons between the same muscles within the same participants.
  • The recovery time between the familiarization session and the assessment session was defined upon references reported in the literature (Courel-Ibáñez et al., 2020). We have included the references in the text.
  • The recovery time between repetitions was selected considering that the participants was adapted to the neuromuscular requirement of the task (Ushiyama et al., 2017; Kraska et al., 2009). We have included the references in the text.
  • We have corrected in the text the explanations in L155-156 and 160-161.
  • Concerning the signal filtering, rectification and smoothing with a low-pass filter is performed with hardware prior to sampling and storing data in the computer, in this way is reduced the bandwidth of the linear envelope (Merletti R. Standard for reporting EMG data. Endorsed by the international Society of Electrophysiology and Kinesiology (ISEK)).
  • The non-gaussian distribution of the data showed a similar trend also after the log transformation, therefore, it was decided to use a more conservative approach (non-parametric statistical procedures).

In our study the dependent variable was the EMG or the combinations of this variable (co-activation index), therefore the multivariate analysis could be inappropriate for the effect of multicollinearity (Thomas JR, 1977).  Additionally, the multivariate analysis is not feasible because our study has a small number of participants (now, the sample size has been reduced).

We have changed in the text all the figures to better highlight the non-gaussian distributions (box plots).         

  • We have corrected the link in the text (http://www.sportsci.org).

Reviewer#2 comment

In the results section:

  • L241. The mean lower limit of ICC is 0.66, not 0.63. In any case, the lower limit occasionally reaches an ICC of 0.10.
  • Some of the lower limits of ICC seem to indicate poor result reproducibility.
  • In section 3.2, the wording of the results is unclear. The figure used combines the representation of different variables and styles (bars vs. points +error bars). Considering the non-normal distribution of the data, it may be preferable to present the results in the form of a table or a figure representing box plots.
  • Effect sizes should also be added in sections 3.2 and 3.3. Additionally, it should be considered to separate the graphs that present the comparisons between the operated and non-operated leg unilaterally and bilaterally, even if they are included in a single image. For example, graphical format A, graphical format B. Otherwise, it appears that the comparisons are made among all groups and tasks.

The discussion is conditioned by the methodological weaknesses previously described, so it is necessary to review it in future stages of the review process.

The conclusion is not clearly aligned with the problem, objective, and results of the study. The results do not allow for the verification of the sensitivity or reliability of the protocol used. The wording of the conclusion needs to be reformulated.

Author’s response

  • Making the ACL group more homogeneous the ICC values improved. The ICC values ranged from “high” to “excellent” in both groups. We have reported the ICC values in supplementary tables (2, 3 and 4) to avoid an excessive of figures and table in the manuscript.
  • The reliability was improved.
  • As reported above we have modified all the figures.
  • The effect sizes are reported for the significant differences.
  • We have modified the figures according to your suggestions.
  • In the discussion section we have included the limitations of our study. We have also expanded the discussion to the CG.

We have modified the conclusions. Anyway, the reliability of unilateral and bilateral exercises was quantified by calculating the ICC values.      

Reviewer 3 Report

Thank you for submitting to Biology.

Overall, a well-organized document.

However, I request Table 1 with the participant's characteristics (including various information such as age, height, weight, athlete career, event distribution, postoperative period, etc.).

It is a well-written literature.

Take a closer look for smooth content flow.

Author Response

Author’s response

First, we would like to thank your comprehensive review of our manuscript. Your comment helped us to improve our study. We have included a Table 1 with the participant's characteristics, in the manuscript. Corrections and changes highlighted with yellow were made in the text.